# Secondary Exfoliation of Electrolytic Graphene Oxide by Ultrasound Assisted Microwave Technique

**DOI:** 10.3390/nano12010068

**Published:** 2021-12-28

**Authors:** Yin Yang, Ziyang Wang, Shaobo Zheng

**Affiliations:** 1State Key Laboratory of Advanced Special Steel & Shanghai Key Laboratory of Advanced Ferro Metallurgy, Shanghai University, Shanghai 200444, China; yinyang@shu.edu.cn (Y.Y.); wzycmx@163.com (Z.W.); 2School of Materials Science and Engineering, Shanghai University, Shanghai 200444, China

**Keywords:** graphene exfoliations, microwave, ultrasonication

## Abstract

Scalable production of large size and high quality graphene is an important prerequisite to fully realize its commercial applications. Herein, we propose a high-efficient route for preparing few-layer graphene. The secondary exfoliation of unexfoliated graphite flakes from electrochemical exfoliation was achieved by using ultrasonication assisted microwave exfoliation technique. The results show that the as-prepared sample has a C/O of 15.2, a thickness of about 1 nm and a transverse dimension of over 100 nm, and the Raman spectrogram shows low defects upon reduction of the sample. These results suggest that electrolytic graphene can be exfoliated to form graphene nanosheets under ultrasonic-assisted microwave technology, thus indicating that the current method has great potential for synthesizing high-quality graphene at an industrial-scale.

## 1. Introduction

Graphene is a carbonaceous material consisting of 2D layered nanomaterials that have a hexagonal dotted structure composed of carbon atoms hybridized with sp^2^ [1]. The peculiarity of graphene is that it has a unique electronic structure and electrical properties that cause it to possess high electron mobility and thermal conductivity. It can also be applied in printed electronics, conductive coatings, composite fillers, solar cells, sensors, lithium-ion batteries, and supercapacitors [2,3,4]. Nanocomposites made of graphene also exhibit many excellent properties. Therefore, it is expected that graphene will have a wide range of applications in the field of materials [5,6,7,8]. Nanocomposites prepared from graphene also exhibit many excellent properties, and it is expected that graphene will have a wide range of applications in the field of materials. The current methods of graphene preparation include mechanical exfoliation, redox, chemical vapor deposition, epitaxial growth, and electrochemical methods [9]. 

Top-down graphene production involves the reduction or exfoliation of powdered graphite, with the most prominent top-down methods being mechanical exfoliation, LPE, electrochemical exfoliation and chemical redox [10,11]. The electrochemical method is based on electrolytic exfoliation under a certain concentration of ionic liquid, which has a low destructiveness on graphite and has attracted special attention due to the simple, rapid, and environmentally friendly nature of electrochemical exfoliation [12]. The secondary exfoliation of unexfoliated graphite flakes from electrochemical exfoliation was achieved by using the ultrasonication assisted microwave exfoliation technique as a typical top-down synthesis method.

Since Geim and Novoselov’s discovery of graphene in 2004 [13], preparing large-area, high-quality graphene has been an important direction for graphene research. Therefore, developing graphene oxide via reduction techniques is currently the main way to prepare graphene in large quantities [14]. Li et al. obtained electrochemically reduced graphene oxide in phosphate buffered solutions at various reduction potentials [15]. Mohandoss et al. studied the reduction of water-dispersed graphene oxide using solar light and illustrated the mechanism of the reduction process [16]. Their study was similar to ours, as both obtained graphene by electrolytically driving anisotropic charged ions and mixtures into graphite electrodes, then swelling and exfoliating the graphite interlayer. The main advantages of this method are as follows: graphite raw materials are abundant and low-cost, have a precise voltage and current regulation, and demonstrate reproducibility and operability during the graphite stripping process [17,18,19].

In recent years, non-chemical reduction has gradually attracted the attention of researchers. Carbon materials are excellent microwave absorbers and can be easily heated by microwave radiation, a property that allows it to be transformed by microwave heating to produce new carbonaceous materials with specific properties [20,21,22,23,24,25,26]. Due to this property of carbon materials, microwave reduction of graphene oxide has become a more suitable alternative reduction method. Zou et al. developed an intermittent microwave exfoliation of a non-expandable graphite oxide process, to prepare graphene with high reduction, high defectivity and high specific surface area, accompanied by excellent supercapacitor performance. The method exponentially improved the efficiency of graphene preparation by redox and successfully solved the two key problems of tedious washing and difficult reduction [21].

Herein, the secondary exfoliation of unexfoliated graphite flakes from electrochemical exfoliation was achieved by using an ultrasonication assisted microwave exfoliation technique. The raw material used is flake graphite, which could not be used directly in our method because of their small size. Thus, they had to be fabricated into macroscopic electrodes to ensure voltage supply. In the post-treatment process, we utilized ultrasonic-assisted microwave technology for the secondary exfoliation of electrolytic graphene. Moreover, we discuss the various characterization results of the experimentally prepared graphene. 

## 2. Materials and Methods

The experiments were conducted by electrochemical exfoliation method to prepare graphene, the graphite powder was briquetted and perforated on the anode titanium rod, and the area not in contact with the graphite block was sealed with wax, and then the graphite block was wrapped using melt-blown cloth to make the anode structure, and the cathode structure was surrounded by the anode using stainless steel ring, and the electrolysis device and schematic diagram of the principle are shown in Figure 1. 

### 2.1. Materials

Materials included: graphite (200 mesh, 99%, AR); H_2_SO_4_ (95~98%, AR); NH_3_·H_2_O (36~38%, AR); HNO_3_ (95~98%, AR); sodium dodecyl sulfate (SDS, total alcohol > 59%); Beaker (2000 mL); titanium rods and; melt-blown cloth (polypropylene material).

### 2.2. Methods

#### 2.2.1. Electrolysis Process

Graphene was prepared by electrolysis, firstly, the beaker was placed on the magnetic heating stirrer, appropriate amount of deionized water was added to the beaker, concentrated nitric acid and concentrated ammonia were added under magnetic stirring, then concentrated sulfuric acid was slowly added, then sodium dodecyl sulfate was added. The specific parameters of the solutions and reaction conditions are shown in Table 1, and finally the electrolyte was fixed to 1000 mL with deionized water. The anode and cathode structures were placed inside the beaker, the DC power supply was turned on, a certain voltage (10 V) were controlled for electrolysis, and the as-prepared electrolysis products were washed and de-dried with distilled water inside the melt-blown cloth repeated several times until the residual substances in the products disappeared, and then the samples were fully dried in a vacuum drying oven at 70 °C to obtain the electrolytic graphene oxide (EGO).

#### 2.2.2. Ultrasound Process

A small amount of the prepared EGO was added to the beaker with the appropriate amount of distilled water. The beaker was then placed in the sonication unit for 15 min and the solution was finally filtered and dried to obtain the sonicated sample.

#### 2.2.3. Microwave Process

The as-prepared EGO was added to the quartz crucible in a small amount, and then the crucible was put into the microwave tube furnace (2.45 Ghz MW-T0316V, Changsha, China), so that the furnace was filled with argon gas and kept in and out of gas equilibrium, under this condition, the microwave power was steadily increased to 800 W by opening the serial port and kept for 60 s, and then the micro wave was turned off and the sample was taken out when the crucible was cooled.

### 2.3. Characterization

The surface morphology of graphene was characterized by a field emission scanning electron microscope (JSM-7500F, Tykyo, Japan) and TEM (JSM-2010F, Tykyo, Japan). The sample thickness was characterized by atomic force microscope (AFM SPM-9600, Tykyo, Japan). Crystalline structure of the as-prepared samples were investigated by powder X-ray diffraction (D8 Advance, Karlsruhe, Germany) at a scan rate of 6° min^−1^. Raman spectroscopy was used for the qualitative characterization of graphene samples. The Raman spectroscopy test equipment was a Horiba LabRam HR Evolution Raman (Madison, WI, USA) spectrometer with a continuous laser at 532 nm as the excitation source.

## 3. Results and Discussion

The chemical composition of the as-prepared EGO was analyzed by combustion elemental analyzer (EA), and shows that EGO has a typical composition (at. %) of C (58.5), O (24.6), H (7.2), N (3.8), and S (5.9). This gives C/O atomic ratio of around 2.4, which indicating a high oxidation degree of EGO. We performed microwave reduction of EGO and found that several-fold expansion occurred in a short time and C/O atomic ratio increased to 15.2 in this process, which was caused by the thermal effect of π-electrons in carbon material inducing a current that propagates in the same phase as the electromagnetic field, resulting in the detachment of oxygen-containing functional groups from the carbon ring. Then, the as-prepared reduced EGO (rEGO) was characterized by AFM, SEM, XRD, and Raman.

### 3.1. Morphological Characterization

We then used an AFM, SEM, and TEM to characterize the number of layers and lateral size of rEGO. AFM can characterize the physical properties of materials including morphology and mechanical properties at the nanometer level, and is unique in terms of layer identification, and SEM has the advantages of nanometer resolution, large observation range and high speed, and is often used to characterize the morphology of grapheme [27,28,29].

The monolayer rEGO sheets show a thickness of about 1.0 nm (Figure 2a) which is due to the penetration of microwaves into the interior of graphite, acting on polar molecules, which generate an electric arc in a flash under the action of a high-frequency electric field with a temperature of up to 1800 °C [24]. High temperatures lead to the decomposition and vaporization of the material inserted between the layers, resulting in axial thrust, so that the graphite laminate structure was opened [25]. It is clear that our prepared samples show a monolayer structure by TEM (Figure 2d,e). We can find that after the electrolytic treatment, a honeycomb structure appears in the longitudinal direction of graphite (Figure 2b), it is an obvious phenomenon of incomplete peeling, but the edges have been well opened, which is due to the decomposition of ions in the solution under the electrolytic action to produce gas between graphite layers, and the force generated by the expansion of gas is greater than the original van der Waals force between graphite layers, which makes the peeling of graphite layers occur [30]. The samples were subjected to ultrasonic and microwave treatment, and it was found that the honeycomb structure of the samples was essentially opened by the high frequency oscillation of ultrasonic waves, and the preliminary lamellar structure appeared (Figure 2c), which had a better peeling effect compared with the experimental results of Grodecki et al. [31].

### 3.2. Structural Characterization

Raman spectroscopy is one of the most commonly used, rapid, non-destructive, and high-resolution techniques to characterize carbon materials. There are three main peaks on the Raman spectrogram of graphene: D, G, and 2D peaks [32]. The D peak generally appears near 1350 cm^−1^ and is caused by the symmetric stretching vibration (radial breathing mode) of the sp^2^ carbon atom in the aromatic ring. It also requires a defect to be activated. The G peak is caused by the stretching vibration between sp^2^ carbon atoms, which corresponds to the vibration of the E_2_g optical phonon in the center of the Brillouin zone. It generally appears near 1580 cm^−1^, while the 2D peak generally appears near 2680 cm^−1^ and is caused by the double resonance leap of two phonons with reverse momentum in the carbon atom [32,33,34]. Its shift, shape, and ratio to the G-peak are closely related to the number of graphene layers. Ferrar et al. showed that the intensity of the G-peak, the intensity ratio of I_G_/I_2D_, and the peak shape of the 2D-peak can determine the number of graphene layers [35].

The graphene oxide prepared by electrolysis has a D-peak (Figure 3). The relative intensity of the D-peak (I_D_/I_G_) represents the degree of defects in EGO, which are caused by the formation of oxygen-containing functional groups on the carbon ring during the electrolysis process. The defects were found to not reduce well after the ultrasonic treatment due to the mechanical vibration that did not remove the oxygen functional groups well. Therefore, we carried out a microwave reduction treatment, during which we used different atmospheres (air, argon, and hydrogen) for the microwave reduction of EGO, as shown in Figure 4, we found that I_D_/I_G_ decreased from 0.92 to 0.11, which means a large reduction of defects in rEGO. And I_G_/I_2D_ decreased from 2.73 to 1.58 and the number of layers of graphene was well controlled. Here, we observed that the relative intensity of the rEGO D-peak after microwaves decreased, corresponding well with EA results, and indicating that ultrasound assisted microwave technique has a good effect on the improvement of EGO defects. 

For graphite structures with more than five layers, it is difficult to distinguish each using Raman spectroscopy [35]. In the Raman spectrum of graphene, single-layer graphene has a very sharp 2D peak with strong symmetry. When there are more than two layers of graphene, the 2D peak half-peak width significantly increases, and the ratio of I_G_/I_2D_ decreases with the decrease of the number of layers. We can find that the ratio of I_G_/I_2D_ decreases in the process of being reduced, which indicates that our reduced EGO has a good quality. This is a considerable improvement in defect and layer control, especially compared to the graphene prepared experimentally by Wang et al. [36].

The crystal structure of the pristine graphite and the as-prepared rEGO were analyzed by conducting X-ray diffraction (XRD) studies (Figure 5). The pristine graphite flakes exhibit a sharp peak centered at 26.36° and 54.54° corresponding to the (0 0 2) and (0 0 4) planes of graphite crystal. EGO after wave radiation treatment shows a distinct peak from the original graphite. The rEGO displays a broad (0 0 2) diffraction peak due to the corrugated structure of the graphene and the stacked graphene layers. The XRD result proved the successful exfoliation of graphite to producing graphene, which is in good agreement with the reported value.

We made the following speculations on the exfoliation process of graphene during the experiments. Under the action of voltage, the ions in the solution enter between the graphite layers and cause the graphite flakes to undergo micro-expansion. During the ultrasonic treatment of the sample, further exfoliation of the graphite interlayer occurs and the ensuing microwave decomposes the graphite interlayer ions and the following decomposition occurs:NO_3_^−^ → NO_2_ + O_2_
SO_4_^−^ → SO_2_ + O_2_
–COOH/–OH → H_2_O + CO_2_ + O_2_

The microwave process, the decomposition of graphite interlayer ions under the effect of high temperature produces a large amount of gas, and the longitudinal thrust generated by the gas is greater than the van der Waals force between the graphite layers, which makes the graphite rapidly expand and separate to form graphene nanosheets. 

## 4. Conclusions

Using elemental analysis, we determined that EGO had a high oxygen content (15.2), which indicates that the graphite was successfully oxidized during the electrolysis process. The oxygen content of the rEGO obtained after the microwave treatment was reduced well, which indicates that the microwave has a good reduction effect on EGO. Through morphological observation and structural characterization, we found that the prepared graphene has excellent quality in terms of layers and defects, and a thickness of about 1 nm and a transverse dimension of over 100 nm. This has lower defects and larger transverse dimensions than the graphene prepared by electrochemical methods by Pei et al. [37].

The above results suggest that EGO synthesized with electrolytic oxidation of graphite powder is conceptually different from Hummers’ method. In this electrochemical method, NH_4_NO_3_ and H_2_SO_4_ mainly act as control agents to tune the anodic electrocatalytic oxygen evolution reaction of water to enable the ultrafast oxidation of graphene lattice. In this process, no other oxidants were used. Therefore, there was no metal ion contamination in this method. The electrolyte was used secondarily because the anode structure is wrapped by melt-blown cloth. Second, in Hummers’ method, graphite oxide is uniformly mixed with concentrated sulfuric acid and other oxidizing agents to form a viscous slurry. The graphite oxide, as a result, is much easier to clean in our method, requiring five times less water. Third, our method is controllable; the degree of oxidation can be adjusted by changing the electrolyte concentration during the oxidation process. Thus, our electrochemical oxidation method combines the advantages of safety, easy control, environmental friendliness, and no metal ion pollution, thus paving the way for low-cost industrial production and application of graphene.

## Figures and Tables

**Figure 1 nanomaterials-12-00068-f001:**
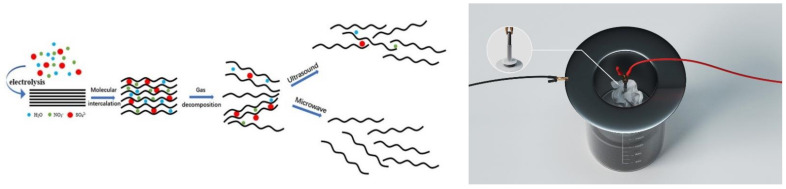
Electrolytic device and schematic diagram for electrochemical preparation of graphene.

**Figure 2 nanomaterials-12-00068-f002:**
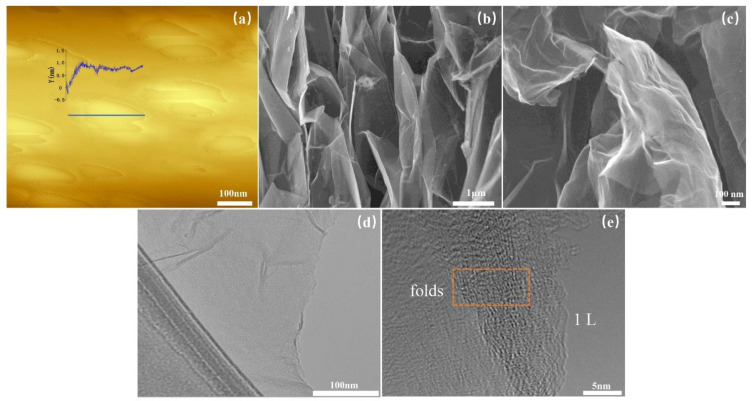
(**a**) rEGO by AFM; comparison of EGO with rEGO, typical SEM (**b**) EGO (**c**) rEGO (**d**,**e**) rEGO by TEM.

**Figure 3 nanomaterials-12-00068-f003:**
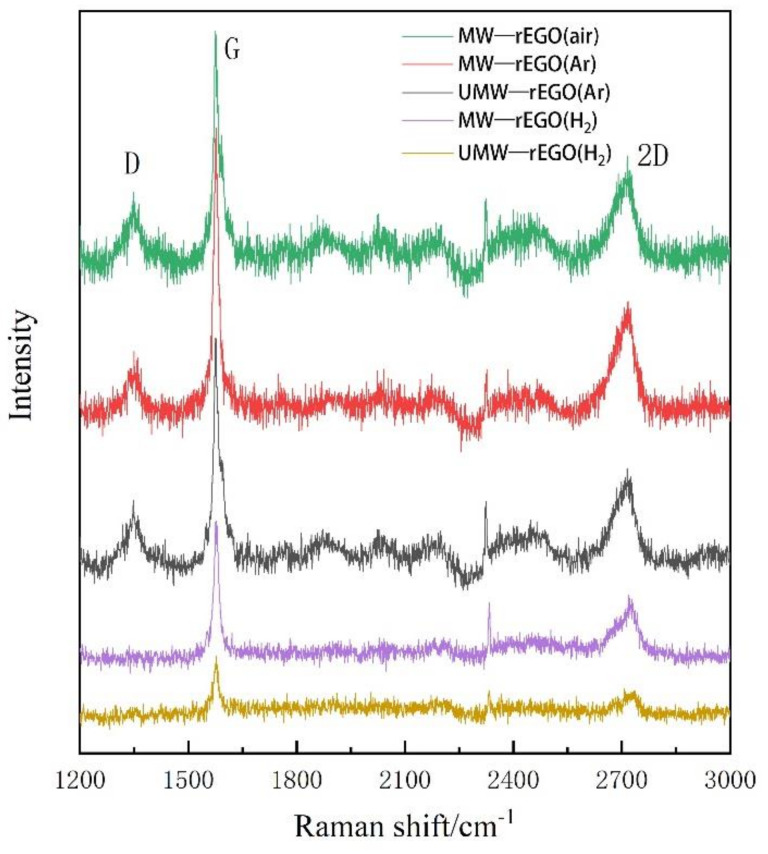
Raman spectra of EGO reduced under different conditions.

**Figure 4 nanomaterials-12-00068-f004:**
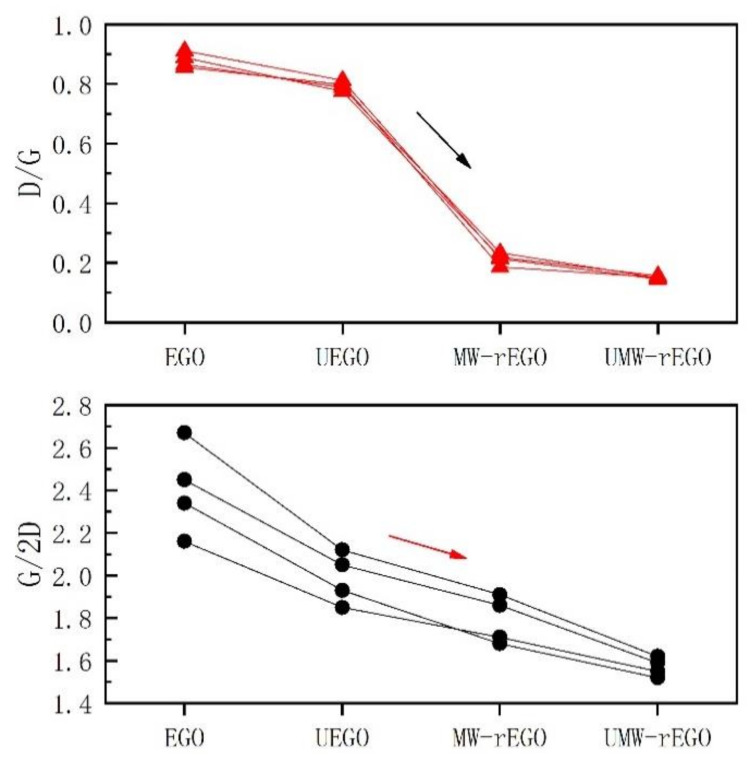
Raman D/G and G/2D of EGO reduced in different ways.

**Figure 5 nanomaterials-12-00068-f005:**
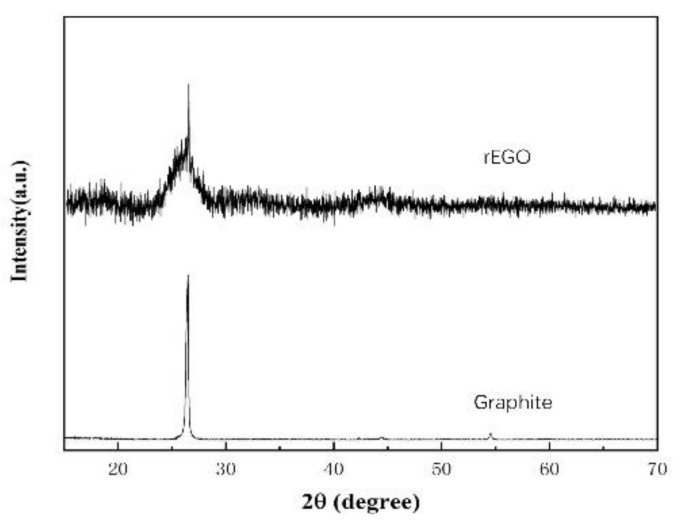
XRD diffraction spectra of graphite flakes and rEGO.

**Table 1 nanomaterials-12-00068-t001:** Electrolyte concentration and subsequent treatment duration.

NH_4_NO_3_ mol/L	H_2_SO_4_ mol/L	SDS mol/L	Electrolysis /min	Ultrasound /min	Microwave /s
4	0.5	0.005	120	15	60

## Data Availability

The data presented in this study are available on request from the corresponding author.

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
