# Peer review of "Secondary Exfoliation of Electrolytic Graphene Oxide by Ultrasound Assisted Microwave Technique"

_nanomaterials, 2021, doi:10.3390/nano12010068_

Round 1

Reviewer 1 Report

 Major revision:

  • Please add more numerical results to the Abstrcat and Conclusion.
  • English throughout the manuscript should be midified.
  • Please add major and minor units to the plots.

Author Response

Dear Reviewer,

Hope this response finds you well.

Thank you very much for your review of my article. After reading your comments on my article, I have taken the following changes.

Query #1: Please add more numerical results to the Abstract and Conclusion.
Our responses:
I have rewritten the abstract section and added relevant numerical results.
The updated text is (Page1): Scalable production of large size and high quality graphene is an important prerequisite to fully realize its commercial applications. Herein, we propose a high-efficient route for preparing few-layer graphene, the secondary exfoliation of unexfoliated graphite flakes from electrochemical exfoliation was achieved by using ultrasonication assisted microwave exfoliation technique. The results show that as-prepared sample has a C/O of 15.2, a thickness of about 1 nm and a transverse dimension of over 100 nm, and the Raman spectrogram shows low defects upon reduction of the sample. Which showed that electrolytic graphene can be exfoliated to form graphene nanosheets under ultrasonic-assisted microwave technology, thus indicating that the current method has great potential for synthesizing high-quality graphene at an industrial-scale.
Query #2: English throughout the manuscript should be midified.
Our responses:
The English of the manuscript has been touched up in MDPI before submission (Edited in a previous submission, not this one) and I have also made some changes to some parts after the return of the manuscript.
Query #3: Please add major and minor units to the plots.
Our responses:
The major and minor scales have been added to Figure 4.
The updated text is (Page5): please check in the attachment.

In addition to the above changes, we have added TEM to give a clearer picture of the number of layers and their quality, and we have added to the introduction of the article to explain the importance of our work. In the conclusion section, we have added a comparison between this work and other relevant recent work to highlight the importance of this work.

Reviewer 2 Report

This manuscript describes exfoliation of graphite by the electrolytic method and reduction with microwave heating and provide characterization data for GO and rGO. However, there are no sufficient data to support their conclusion. And, I cannot find any advantages of this method comparing with previously established electrolytic exfoliation methods of graphite.

1)The chemical structure of graphene nanosheets must be characterized by XPS or 13C NMR to clearly define their structure of GO and rGO.

2) The AFM image in Figure 2 do not provide the clear GO image showing their lateral size. Form this, I am not sure that the produced GO is a single layer sheet. It would be better to provide TEM image showing crystalline structures.

3) In Figure 3, all of Raman spectra are not from GO but rGO. From Raman spectra, asymmetric 2D peak indicates that produced rGO is not a single layer sheet, which must be supported by other data. It should be better to insert Raman spectrum of graphite in Figure 3. And the plot should be normalized by G-band to see the difference of D-band and 2D-band peaks.

4) What about the DC power used in exfoliation of graphite?

5) The font of the first paragraph of Results section is different.

Author Response

Dear Reviewer,

Hope this response finds you well.

Thank you very much for your review of my article. After reading your comments on my article, I have taken the following changes.

Query #1: The chemical structure of graphene nanosheets must be characterized by XPS or 13C NMR to clearly define their structure of GO and rGO.
Our responses:
The XPS is a good way to characterize the carbon skeleton structure. For this article I did not do an XPS due to atual equipment reasons, but did an elemental analysis. The result of elemental analysis means that there are a large number of oxygen-containing functional groups in the carbon skeleton, and the reduction in oxygen content after reduction means that these oxygen-containing functional groups are effectively removed.
Query #2: The AFM image in Figure 2 do not provide the clear GO image showing their lateral size. Form this, I am not sure that the produced GO is a single layer sheet. It would be better to provide TEM image showing crystalline structures.
Our responses:
For the question of whether a single layer of graphene has been produced, I did a TEM test to clearly show the crystal structure.
The updated text is (Page4): please check the attachment - Figure 2. (a)rEGO by AFM; Comparison of EGO with rEGO ,Typical SEM (b) EGO (c) rEGO (d)(e) rEGO by TEM

The monolayer rEGO sheets show a thickness of about 1.0 nm (Fig. 2a) which is due to the penetration of microwaves into the interior of graphite, acting on polar molecules, which generate an electric arc in a flash under the action of a high-frequency electric field with a temperature of up to 1800℃, [24]. High temperatures lead to the decom-position and vaporisation of the material inserted between the layers, resulting in axial thrust, so that the graphite laminate structure was opened[25]. interlayer inserted material decomposition vaporization, resulting in axial thrust, so that the graphite laminate structure was opened [24]. It is clear that our prepared samples show a mon-olayer structure by TEM(Fig2. d, e).
Query #3: In Figure 3, all of Raman spectra are not from GO but rGO. From Raman spectra, asymmetric 2D peak indicates that produced rGO is not a single layer sheet, which must be supported by other data. It should be better to insert Raman spectrum of graphite in Figure 3. And the plot should be normalized by G-band to see the difference of D-band and 2D-band
peaks.
Our responses:
The peak shape of the two-dimensional peaks in the Raman pattern can in some sense indicate the number of layers of graphene, but it is difficult to determine the symmetry of more than two layers of graphene, as has been mentioned in previous literature (Ferrari, A. C. Solid State Communications 143, 47-57). I have added a discussion of the ratio of Raman peaks in the paper to determine the quality of graphene defects by the rise and fall of the ratio. The production of graphene using electrolytic graphene powder is somewhat difficult to guarantee the homogeneity of the quality, which also has some uncertainty for Raman detection, a great challenge for the current large-scale application of graphene.
The updated text is (Page6):
The defects were found to not reduce well after the ultrasonic treatment due to the mechanical vibration that did not remove the oxygen functional groups well. Therefore, we performed a microwave reduction treatment, during which we used different atmospheres (air, argon and hydrogen) for the microwave reduction of EGO, as shown in Figure 4, we found that ID/IG decreased from 0.92 to 0.11, which means a large re-duction of defects in rEGO. And IG/I2D decreased from 2.73 to 1.58 and the number of layers of graphene was well controlled. during the experiment, we used different atmospheres (air, argon and hydrogen) for the microwave reduction of EGO and found that the D peak of EGO almost disappeared in the environment of hydrogen, which indicated that hydrogen was the best for the reduction of EGO. Here, we observed that the relative intensity of the rEGO D-peak after microwaves decreased corresponded well with EA results, indicating that ultrasound assisted microwave technique has a good effect on the improvement of EGO defects.
Query #4: What about the DC power used in exfoliation of graphite?
Our responses:
The DC voltage used in the electrolysis process was 10V, which has been added in the paper.
Query #5: The font of the first paragraph of Results section is different.
Our responses:
Some of the font differences in the results are due to my own error and have been corrected.
In addition to the above changes, the abstract and introduction sections of the article have been added to explain the importance of our work. A comparison of this work with other relevant recent work has been added to the conclusion section to highlight the importance of this work.

Reviewer 3 Report

Section

Comments

Title

The title looks good

Abstract section

·        The abstract seems very poor with important result that are mostly need to be reported , such as the properties of the exfoliated graphene ( number of layers, amount of defects, electrical conductivity, C/O ratio .etc.)

·        My recommendation is to rewrite the abstract and try to list the most important findings from this work by which the reader can read it individually and identify the significance of the article.

Keywords

·        What’s about editing the keywords as follows:

“ graphene exfoliations ; microwave ; ultrasonication and remove the characterization keywords which can be replaced with appropriate related words.

Introduction section

·        Lines 31 – 36, the authors mixed between top-down exfoliation methods and bottom-up methods. Then, from line 32 -36 , the authors focused on the electrochemical exfoliation method. Please, describe briefly the top-down approaches since the ultrasound assisted microwave technique is within the top-down “liquid phase exfoliation methods”. Therefore, it is useful to talk briefly about the top-down exfoliation methods and focus on the ultrasonication assisted microwave technique. I recommend to go through the most recent published article on the graphene exfoliation method which discussed each method intensively which can be read on the following link:
https://www.sciencedirect.com/science/article/pii/S2213343721014834#!

·         Lines 51 -54 , supported reference is required.

·        Lines 62-68, the contribution of this work paragraph need to be modified , for example:
“ Herein, the secondary exfoliation of unexfoliated graphite flakes from electrochemical exfoliation was achieved by using ultrasonication assisted microwave exfoliation technique …..”

Section 2.

·          

Materials and Methods

2.2.1    Electrolysis process:

·        In line 90, please mention the amount of voltage applied.

2.2.2. Ultrasound process:

·        This section needs to be improved regarding to the English language. Please, rewrite in passive voice.

2.3. Characterization:

Lines 111-112 , the XRD equipment module and specification needs to be described.

Section 3

Results and Discussion:

 3.1. Morphological characterization:

·        In lines 134- 138,  the sentence was very long and need to be divided into two sentences with inserting the supporting references.

·        In lines 138 – 144, the same note as previous one. The sentence is too long and need to be divided into two sentences with adding the relevant references to support the findings.

·        Please, modify the reported ID/IG and I2D/IG ratios abbreviations like this that can be seen elsewhere: (ID/I;  I2D/IG)

3.2. Structural characterization:

·        Lines 158 – 161, supporting references are required .

·        Lines 181-186, it is hard to consider the graphene has few layers or high quality from Raman diffraction peaks. So, it is important to perform HRTEM test for the exfoliated graphene in this work to know more about the number of layers and its quality.

·        The calculation of ID/IG and I2D/IG is necessary to be added in this section especially when the authors demonstrated the amount of  defects.

Section 4

 4. Conclusion :

·        Please, move the text from line 206-218 to the discussion section. The mechanism of the intercalated ions should be not reported in the conclusion section.

·        A comparison table between this work and other related recent work should be added  before the conclusion section in order to highlight the significance of this reported article.

Author Response

Dear Reviewer,

Hope this response finds you well.

Thank you very much for your review of my article. After reading your comments on my article, I have taken the following changes.

Query #1: The abstract seems very poor with important result that are mostly need to be reported , such as the properties of the exfoliated graphene ( number of layers, amount of defects, electrical conductivity, C/O ratio .etc.). My recommendation is to rewrite the abstract and try to list the most important findings from this work by which the reader can read it individually and identify the significance of the article.
Our responses:
The abstract section does not give much prominence to the properties of graphene and has been rewritten with your suggestions in mind to demonstrate the importance of the article. Also keywords have been replaced and removed as appropriate.
The updated text is (Page1): Scalable production of large size and high quality graphene is an important prerequisite to fully realize its commercial applications. Herein, we propose a high-efficient route for preparing few-layer graphene, the secondary exfoliation of unexfoliated graphite flakes from electrochemical exfoliation was achieved by using ultrasonication assisted microwave exfoliation technique. The results show that as-prepared sample has a C/O of 15.2, a thickness of about 1 nm and a transverse dimension of over 100 nm, and the Raman spectrogram shows low defects upon reduction of the sample. Which showed that electrolytic graphene can be exfoliated to form graphene nanosheets under ultrasonic-assisted microwave technology, thus indicating that the current method has great potential for synthesizing high-quality graphene at an industrial-scale. Keywords: Graphene exfoliations; Microwave ; Ultrasonication
Query #3: Lines 31 – 36, the authors mixed between top-down exfoliation methods and bottom-up methods. Then, from line 32 -36 , the authors focused on the electrochemical exfoliation method. Please, describe briefly the top-down approaches since the ultrasound assisted microwave technique is within the top-down “liquid phase exfoliation methods”. Therefore, it is useful to talk briefly about the top-down exfoliation methods and focus on the ultrasonication assisted microwave technique. I recommend to go through the most recent published article on the graphene exfoliation method which discussed each method intensively which can be read on the following link:
https://www.sciencedirect.com/science/article/pii/S2213343721014834#!
Our responses:
In the introduction section I read relevant articles based on recommendations and then briefly described the top-down approach to graphene preparation.
The updated text is (Page1): Top-down graphene production involves the reduction or exfoliation of powdered graphite, with the most prominent top-down methods being mechanical exfoliation, LPE, electrochemical exfoliation and chemical redox[10, 11]. The electrochemical method is based on electrolytic exfoliation under a certain concentration of ionic liquid, which has a low destructiveness on graphite and has attracted special attention due to the simple, rapid, and environmentally friendly nature of electrochemical exfoliation [12]. The secondary exfoliation of unexfoliated graphite flakes from electrochemical exfoliation was achieved by using ultrasonication assisted microwave exfoliation technique is a typical top-down synthesis method.
Query #4: Lines 51 -54 , supported reference is required. Lines 62-68, the contribution of this work paragraph need to be modified , for example: “ Herein, the secondary exfoliation of unexfoliated graphite flakes from electrochemical exfoliation was achieved by using ultrasonication assisted microwave exfoliation technique …..”
Our responses:
The overly long phrases in the article have been regrouped and appropriately quoted.And The statements in the contribution section of this work have been modified.
The updated text is (Page2):
In recent years, many non-chemical reduction has gradually attracted the atten-tion of researchers. Carbon materials are very good microwave absorbers and can be easily heated by microwave radiation, a property that allows it to be transformed by microwave heating to produce new carbonaceous materials with specific properties[20-26]. Herein, the secondary exfoliation of unexfoliated graphite flakes from electro-chemical exfoliation was achieved by using ultrasonication assisted microwave exfoli-ation technique. The raw material used is flake graphite, Which could not be used di-rectly in our method because of their small size……
Query #4: In line 90, please mention the amount of voltage applied.
Our responses:
The voltage of 10 V during the electrolysis of this experiment has been added.
The updated text is (Page3): …the DC power supply was turned on, a certain voltage(10V) were controlled for electrolysis…
Query #5: This section needs to be improved regarding to the English language. Please, rewrite in passive voice.
Our responses:
Language in 2.2.2 has been changed with passive voice.
The updated text is (Page3): A small amount of the prepared EGO was added to the beaker with the appropri-ate amount of distilled water. The beaker was then placed in the sonication unit for 15 minutes and the solution was finally filtered and dried to obtain the sonicated sample.
Query #6: Lines 111-112 , the XRD equipment module and specification needs to be described.
Our responses:
the model number of the XRD is D8 Advance, which has been added in full in the article.
Query #7: Morphological characterization:
In lines 134- 138, the sentence was very long and need to be divided into two sentences with inserting the supporting references.
In lines 138 – 144, the same note as previous one. The sentence is too long and need to be divided into two sentences with adding the relevant references to support the findings.
Please, modify the reported ID/IG and I2D/IG ratios abbreviations like this that can be seen elsewhere: (ID/IG ; I2D/IG)
Our responses:
The overly long phrases in the article have been regrouped and appropriately quoted.And ID/IG and I2D/IG ratios have been made in the text appropriately.
The updated text is (Page4):
The monolayer rEGO sheets show a thickness of about 1.0 nm (Fig. 2a) which is due to the penetration of microwaves into the interior of graphite, acting on polar molecules, which generate an electric arc in a flash under the action of a high-frequency electric field with a temperature of up to 1800℃ [24]. High temperatures lead to the decompo-sition and vaporisation of the material inserted between the layers, resulting in axial thrust, so that the graphite laminate structure was opened[25].
Query #8: Structural characterization:
Lines 158 – 161, supporting references are required .
Lines 181-186, it is hard to consider the graphene has few layers or high quality from Raman diffraction peaks. So, it is important to perform HRTEM test for the exfoliated graphene in this work to know more about the number of layers and its quality.
The calculation of ID/IG and I2D/IG is necessary to be added in this section especially when the authors demonstrated the amount of defects.
Our responses:
We have introduced relevant literature in designated places to better support this; For the question of whether a single layer of graphene has been produced, we did a TEM test to clearly show the crystal structure. The discussion of peak ratios for Raman has been added to the text.
The updated text is :
The D peak generally appears near 1350 cm-1 and is caused by the symmetric stretch-ing vibration (radial breathing mode) of the sp2 carbon atom in the aromatic ring, it also requires a defect to be activated [33]. The G peak is caused by the stretching vibra-tion between sp2 carbon atoms, which corresponds to the vibration of the E2g optical phonon in the center of the Brillouin zone [34]. It generally appears near 1580 cm-1, while the 2D peak generally appears near 2680 cm-1 and is caused by the double reso-nance leap of two phonons with reverse momentum in the carbon atom [32, 35, 36].

Please check the attachment: Figure 2. (a)rEGO by AFM; Comparison of EGO with rEGO ,Typical SEM (b) EGO (c) rEGO (d)(e) rEGO by TEM
The monolayer rEGO sheets show a thickness of about 1.0 nm (Fig. 2a) which is due to
the penetration of microwaves into the interior of graphite, acting on polar molecules, which generate an electric arc in a flash under the action of a high-frequency electric field with a temperature of up to 1800℃, [24]. High temperatures lead to the decom-position and vaporisation of the material inserted between the layers, resulting in axial thrust, so that the graphite laminate structure was opened[25]. interlayer inserted material decomposition vaporization, resulting in axial thrust, so that the graphite laminate structure was opened [24]. It is clear that our prepared samples show a mon-olayer structure by TEM (Fig2. d, e).
The defects were found to not reduce well after the ultrasonic treatment due to the mechanical vibration that did not remove the oxygen functional groups well. Therefore, we performed a microwave reduction treatment, during which we used different atmospheres (air, argon and hydrogen) for the microwave reduction of EGO, as shown in Figure 4, we found that ID/IG decreased from 0.92 to 0.11, which means a large re-duction of defects in rEGO. And IG/I2D decreased from 2.73 to 1.58 and the number of layers of graphene was well controlled.
Query #9: Conclusion :
Please, move the text from line 206-218 to the discussion section. The mechanism of the intercalated ions should be not reported in the conclusion section.
A comparison table between this work and other related recent work should be added before the conclusion section in order to highlight the significance of this reported article.
Our responses:
The conclusion section has also been suitably revised in response. The part of mechanism of the intercalated ions has been moved to the results section. In the conclusion section, we have added comparisons of previous experiments to highlight our importance.
The updated text is (Page7): Through morphological observation and structural characterization, we found that the prepared graphene has excellent quality in terms of layers and defects and a thickness of about 1 nm and a transverse dimension of over 100 nm. This has lower defects and larger transverse dimensions than the graphene prepared by electrochemical methods by Pei et al[39].

Round 2

Reviewer 1 Report

The comments have been done.

It can be accepted for publication.

Reviewer 3 Report

Thank you very much for your serious consideration of my comments, and for having given the response. Meanwhile, the manuscript has been greatly revised to incorporate these comments. I think the current version could be published in Nanomaterials Journal . Please double-check any possible errors in the manuscript before publication to avoid regrets. Looking forward to read more of your research results in the future. Thank you again for your careful response to these comments and for your efforts to improve the quality of the manuscript.